# Understanding the Role of Leisure in Portuguese Adolescent Wellbeing Experience

**Linda Caldwell [1],\*** and **Teresa Freire [2]**

[1] Department of Recreation, Park, and Tourism Management, Pennsylvania State University, University Park, PA 16802, USA

[2] Department of Applied Psychology, Psychology Research Centre, University of Minho, 4710-057 Braga, Portugal

\* Correspondence: lindac@psu.edu

**Abstract:** Because adolescent leisure is important to development, we consider its role in Portuguese adolescent wellbeing. Data for this study came from 303 adolescents in grades 10, 11, and 12 living in a large urban area in northern Portugal. Self-report data were collected in classrooms using a cross-sectional design in two urban high schools. Hypothesis testing used seven hierarchical linear regression models. Except for subjective happiness, experiencing boredom in leisure and/or the ability to make a boring situation more interesting were strong predictors of each wellbeing experience in the predicted direction. Perceptions of healthy leisure were associated with higher levels of life satisfaction, subjective happiness, self-esteem, and positive affect. Active leisure was important to adolescent self efficacy and positive affect. Those who could restructure a boring situation into something more interesting exhibited higher levels of wellbeing experience. Adolescents who perceived parental autonomy control were more likely to experience boredom in leisure.

**Keywords:** boredom; leisure; parent autonomy control; parent autonomy support; self-determination; wellbeing experience

## 1. Introduction

As introduced by early scholars, such as Aristotle and Socrates, leisure is considered essential for the body, mind, and soul—in short, optimal wellbeing. In contemporary times, leisure is viewed as an important developmental context in which the lives of youth can be significantly improved [1–3]. However, leisure can also be fraught with social and personal problems, such as unhealthy and potentially risky behavior (e.g., substance abuse) [4]. From a positive youth development perspective, it is important to understand factors that increase or decrease positive life experiences, promote potentialities, and prevent risks. Because adolescents spend much time in leisure activities, we consider the role of leisure in Portuguese adolescent wellbeing.

The study of adolescent wellbeing is fundamental to understanding lifestyle choices and their influence on developmental trajectories [5]. This perspective focuses on personal and ecological conditions that promote and support adolescents in learning to manage developmental experiences and opportunities. Learning to manage these experiences contributes to adolescents directly influencing their own development [6] in order to develop fulfillment, capability, and a positive sense of self [7]; these can be expressed in terms of wellbeing. Adolescent wellbeing is important because it appears to be a predictor of better health and less risky behaviors in adulthood [8,9].

### 1.1. Wellbeing Experience

Two perspectives, subjective well-being (hedonic perspective) and psychological well-being (eudaimonic perspective), contribute to the understanding of general wellbeing in terms of life satisfaction, happiness, meaning, and personal growth [10,11]. Subjective

wellbeing refers to how individuals evaluate and experience their lives in positive (vs. negative) ways, including a global subjective judgment about one's life overall and the frequency of one's positive and negative affective experiences [12,13]. Psychological wellbeing focuses on growing and developing as a person, pursuing meaningful goals and values, and coping with life challenges [14,15]. In this paper, subjective and psychological wellbeing are integrated under the term wellbeing experience (WBE) to refer to the general experience of wellbeing that, according to the literature, comprises several relationships with other concepts, such as subjective happiness and life satisfaction, self-esteem and self-efficacy, and positive and negative affect.

Specifically, life satisfaction is a judgment about one's overall life and is considered the cognitive component of subjective wellbeing [16]. It is based on how people believe their life should be in relation to how it is [17,18]. Subjective happiness is a global subjective assessment about being a happy or unhappy person, regardless of one's meaning or perceived source of happiness [19]. In adolescence, greater life satisfaction has been associated with subjective wellbeing [20–23] and negatively related to psychological distress among Portuguese adolescents [8].

Like life satisfaction and subjective happiness, self-esteem and general self-efficacy are important to self-regulatory processes [24]. Self-esteem reflects the positivity of one's self-evaluations [25,26] and is an affective evaluation of general feelings about oneself, self-worth, and value [24]. It is a basic domain of human functioning, important for social interaction, mental health, and wellbeing [27]. General self-efficacy refers to a stable sense of personal competence to deal effectively with a variety of stressful situations, reflecting how efficacious people judge themselves across various domains of functioning [28,29]. In adolescents, self-esteem [30] and general self-efficacy [30–32] are positively related with subjective wellbeing [32,33].

The affective or emotional component of self-evaluations is also related to adolescent WBE with an emphasis on positive and negative affect. Positive and negative affect are distinct dimensions of affective structure with negative expressing general feelings of distress and displeasure [33]. Negative affect is a risk factor for psychological and adjustment problems. Positive affect is best understood as liveliness and pleasurable engagement with one's environment and may serve as a protective factor against maladaptive outcomes [34]. Low positive affect is characterized by the absence of these feelings, reflecting lethargy or sadness [33].

These WBE variables are well supported in the literature and reflect dimensions of positive adolescent development. Furthermore, the positive role leisure plays in adolescent development, health promotion, and risk prevention has grown over the past two decades. However, little research has been conducted that examines the role of leisure with regard to these WBE variables, although one study has examined the role specific personality factors in contrast to type of leisure activity played in adolescent psychological wellbeing [35]. No research has been conducted in the cultural context of Portuguese adolescents.

### 1.2. Adolescent Development and Leisure

The Leisure, Activities, Context, and Experience model (LACE [36]) suggests what adolescents do in their free or leisure time (activities), where they do it and the supports or opportunities that are available (context), and how they experience what they do (experience) combine to help explain developmental and wellbeing outcomes associated with leisure participation. When youth engage in what are considered leisure or recreational activities in their free time, the assumption is that they have freely chosen to participate, and, thus, experience a high degree of perceived freedom, have a level of personal control over the activity, and generally have a positive experience.

Leisure is self-determined and connected to perceived freedom and perceived control which differentiates it from activities conducted in other developmental contexts, such as school or work. These latter activities are typically associated with lower levels of self-determination, personal freedom, and personal control despite often producing pos-

itive developmental and wellbeing outcomes. Thus, leisure is considered an important developmental context [37].

The study of healthy leisure has gained momentum as it is considered a protective factor against risky behavior (e.g., [38]). Characteristics of healthy leisure include being actively engaged in meaningful and/or creative activities, being optimistic, positive environments, and positive social experiences [39].

In addition to healthy leisure, the literature suggests the experience of boredom during leisure is associated with negative outcomes, such as substance use and sexual risk (e.g., [40,41]), yet we are unaware of any studies that examine how boredom in leisure is associated with happiness and wellbeing. Theoretically, being bored should motivate a person to do something to alleviate boredom and turn the situation into something more interesting [1]. The ability to self regulate in this way is important because the inability to restructure a boring situation is associated with higher levels of substance use among South African adolescents between grades 8 and 10 [41,42]. Therefore, adolescents who can restructure a boring situation into something more interesting may well reap positive wellbeing benefits as compared to their peers who do not have that skill.

There is a vast literature on the role of leisure activity participation, and in particular structured or organized activity participation, on positive development (e.g., [1,42]). Hallmarks of these types of activities, which include youth clubs, sports, and cultural or artistic endeavors, is that they provide a context for active engagement with one's environment (e.g., [43]). Research suggests that youth who spend more hours engaged in active leisure pursuits have higher levels of wellbeing, and logically, would be less bored. More passive activities such as socializing, relaxing, internet and entertainment activities are less likely to produce positive outcomes, although they are still important to overall wellbeing.

The role of parents in their adolescents' leisure lives is important in facilitating positive growth and mitigating risky behaviors, such as substance use and sexually risky behavior [44,45]. As adolescents mature, the balance between parental authority and adolescent autonomy is more likely to be strained in the relatively free leisure context that offers a fertile time for exploration and experimentation; the development of autonomy, identity, and initiative; and the opportunity to make mistakes as well as take personal responsibility. The literature suggests adolescents who perceive their parents to be autonomy-supporting feel more self determined, whereas those who feel parents are autonomy-= controlling are more likely to feel amotivated [46] and a loss of self determination [47–49]. Especially in leisure, it is important for adolescents to have the opportunity to experience the self determination necessary for an optimal leisure experience. For example, adolescents who perceive their parents to be autonomy controlling tend to experience higher levels of boredom in leisure [45].

The role of parents is particularly important in Portugal, as family is more important than employment, leisure, friends, politics, and religion [50,51]. Parental control of adolescent internet use, for example, is related to lower levels of internet addiction among a sample of Portuguese youth [52]. Communication with parents is also linked to quality of life among Portuguese families [51]. Analyzing data from Health Behavior in School-aged Children/HBSC [53] researchers found that adolescents who reported having a lower quality of life reported difficulties in communication with both mothers and fathers, while those who reported their parents helped with decisionmaking and treated them with fairness overwhelming reported higher levels of quality of life [51].

In this study, we included three aspects of perceived parental influence: parental knowledge of what their children do (knowledge), the degree to which parents help their children participate in leisure activities (autonomy support), and whether their involvement is considered overly controlling (autonomy control). It is worth noting that we did not measure adolescents' disclosure to their parents about what they do (e.g., [49]), only their perceptions of the role of their parents in their leisure pursuits.

### 1.3. The Context

In this study we focus on 10th to 12th grade Portuguese adolescents. In terms of general wellbeing and population characteristics, we turned to research based on the nationally representative Portuguese 2009/10 Health Behavior in School-Aged Children Study [53,54]. The survey indicated that about a quarter of those sampled were overweight or obese. Close to 57% of those surveyed reported good family communication. Having been bullied was relatively rare, although 37% reported being bullied once or more. A little over half of youth reported being physically active for 60 min on three or more days a week, although only about 13 percent met the international physical activity recommendation of one hour per day. Portuguese girls are the most inactive in Europe. About a quarter of respondents reported using screen time for five or more hours per day. Close to 90% of youth reported not using tobacco products, 10% reported having drunk alcohol more than twice over the past 30 days, about 93% report never having been drunk, and about 98% reported never using cannabis.

The Portuguese National Mental Health Plan (PNSM) [55] put forth a number of priority areas to address child and adolescent health, starting in 2020. This priority is partly in response to the fact that suicide rates are the leading cause of death among adolescents between the ages of 15 and 24. In a related effort, researchers at the Higher School of Health of the Polytechnic Institute of Viseu undertook a study to better understand how to prevent mental disorders and promote positive mental health in the school system [56,57]. These efforts are based on an ecological framework that educates and empowers those who work with young people to develop programs and policies that promote wellbeing, recognizing the interaction between mental health and quality of life.

The Global Youth Development Index and Report [58] indicates that among the 42 countries that contributed to the report on youth aged 15 to 19 years; Portugal ranked 9th in terms of the highest achievement for youth development. This report also acknowledges the role leisure plays in adolescent development and wellbeing. In Portugal, however, leisure remains a nascent area of inquiry, particularly regarding adolescent lives. This growing area of interest includes a focus on adolescent leisure experience from a positive youth development perspective, e.g., [59,60].

Our study was situated in an area in northern Portugal that is about one hour's drive to Porto (the second biggest city after Lisbon) which is on the coast. The area was established over two thousand years ago by the Romans and is home to the oldest city in Portugal. It has a mild climate and abundant natural resources as it is located near the southern border of Penêda-Gerês National Park. The municipality/city has approximately 200,000 people and coupled with the surrounding region, it is the third most urban and densely populated city in Portugal. Youth under the age of 25 constitute 35% of the population and the majority (approximately 98%) of the population is native Portuguese with those from Brazil the second highest proportion. In 2012, the municipality was named the European Youth Capital, a designation that signifies a city that supports its youth through projects designed to promote participation in cultural, political, social, and economic endeavors to improve living conditions. Residents are primarily of the Catholic faith and there are many significant churches and monasteries in the municipality.

### 1.4. The Current Study

The primary purpose of this study is to examine the role of leisure in adolescent wellbeing experience among a sample of Portuguese high school students. The hypotheses include:

1. Participating in active leisure, healthy leisure, and the ability to restructure a boring situation in leisure (independent variables) will positively predict these dependent variables: life satisfaction, subjective happiness, self-esteem, general self-efficacy, and positive affect. Being bored in leisure will negatively predict those dependent variables.
2. Participating in active leisure, healthy leisure, and the ability to restructure a boring situation in leisure (independent variables) will negatively predict negative affect (dependent variable). Being bored in leisure will positively predict negative affect.

The secondary purpose of this paper is to examine predictors of boredom in leisure. Our hypotheses are:

3. Positive perceptions of parental autonomy control (independent variable) will positively predict boredom (dependent variable).
4. Positive perceptions of parental knowledge and autonomy support, healthy leisure, and ability to restructure a boring situation in leisure will negatively predict boredom in leisure.

## 2. Materials and Methods

### 2.1. Sample and Procedure

Data for this study came from 303 adolescents living in large urban area in northern Portugal. Of the 303 youth who participated in the study, the average age is 16, and about 59% are female. Forty-three percent are in grade 10, 35% in grade 11, and 17% in grade 12. The majority (81%) live in the central city and the others live in the city district but outside of the central city.

Using a cross-sectional design and convenience sampling, and after gaining permission from school directors and appropriate teachers, we collected data in two main city high schools. Research assistants recruited participants and collected data in classrooms after informing students about the aims of the study, explaining that their participation was voluntary and answers were anonymous, and how to access the self-report, online survey via either mobile devices or computers provided in the schools. After students gave their informed consent, they were given access to the items on the survey. Participants generally completed the questionnaire within 15 min. This study was approved by the Ethics Committee for Research in Social and Human Sciences (CEICSH)—Project number CEICSH 046/2019.

### 2.2. Measures

The Subjective Happiness Scale [19,61]; Portuguese adolescent validation is a 4-item measure using a 7-point Likert-type response scale (1 = lowest level of happiness to 7 = highest level of happiness). In two items, individuals characterize themselves by absolute and relative comparison with their peers (e.g., "In general, I consider myself: not a very happy person versus a very happy person") while in the other two items, they indicate the extent to which they identify in general with happy and unhappy people. In the present study, $M = 4.70$ ($SD = 0.83$) and Cronbach's alpha = 0.83.

Satisfaction with Life Scale [62,63]; Portuguese adolescent validation] assesses global cognitive judgments of satisfaction with life in general (e.g., "In most ways, my life is close to my ideal") with five items using a 7-point Likert-type response scale (1 = strongly disagree to 7 = strongly agree). In the present study, $M = 4.56$ ($SD = 1.29$) and Cronbach's alpha = 0.83.

The Rosenberg Self-Esteem Scale [64,65]; Portuguese adolescent validation is a 10-item measure using a 4-point Likert-type response scale (1 = strongly disagree to 4 = strongly agree), with five items being reversed. It assesses general feelings of self-esteem, or the favorable/unfavorable attitude toward oneself (e.g., "On the whole, I am satisfied with myself"). In the present study, the summed score = 27.46 ($SD = 5.72$) and Cronbach's alpha = 0.88.

The General Self-efficacy Scale ([29,66]; Portuguese adolescent validation) is a 10-item multidimensional scale using a 10-point Likert-type response scale (1 = not at all like me to 10 = totally like me) and assesses the strength of an individual's belief in their own ability to respond to novel or difficult situations and to deal with any associated obstacle or setbacks (e.g., "I can always manage to solve difficult problems if I try hard enough"). In the present study, the summed score was 66.77 ($SD = 15.67$) and Cronbach's alpha = 0.90.

PANAS-VRP ([33,67] Portuguese validation, short version) has ten items that address emotions in life in general divided into two independent dimensions, positive affect (5 items) and negative affect (5 items), using a 5-point Likert-type response scale (1 = nothing or very little to 5 = extremely). Higher scores on negative affect indicate displeasure and subjective discomfort whereas positive affect indicates pleasure and subjective wellbeing.

In the present study, Cronbach's alpha = 0.79 for positive affect (*M* = 3.59, *SD* = 0.79) and 0.84 for negative affect (*M* = 2.47, *SD* = 0.89).

The leisure-related variables were measured on a five-point Likert-type scale where 1 = strongly disagree and 5 = strongly agree, except for active leisure which had a different response scale. The boredom in leisure and active leisure measures were specifically designed for this study in Portugal; other measures have been used in previous studies (e.g., [4,41]). Characteristics of the measure include:

Ability to restructure a boring leisure situation was measured by six items (e.g., "I know how to turn a boring situation into something that is more interesting to me"); *M* = 3.63 (*SD* = 0.62) and Cronbach's alpha = 0.81.

Healthy leisure was measured using five items (e.g., "The things that I do in my free time are healthy".) The mean score was 3.52 (*SD* = 0.73) and Cronbach's alpha = 0.77.

Active leisure is a combination of four activities that typically require active engagement, such as participating in sports or physical activity, or outdoor activities, such as hiking. There were four response options that reflected how many hours per week the adolescent participated in the activity over the past six months: 1 = 1–3 h, 2 = 4–6 h, 4 = 7–9 h, and 5 = 10 or more hours. The mean score was 2.01 (*SD* = 0.61).

Perceived parental knowledge was measured with three items, *M* = 3.89 (*SD* = 0.76) and Cronbach's alpha = 0.70. A sample item is "My parents know a lot about me and what I enjoy most in my free time".

Three items measured perceived parental autonomy support; *M* = 3.41 (*SD* = 0.88) and Cronbach's alpha = 0.79. A sample item is "My parents help me make decisions in my free time".

Perceived parental autonomy control was measured with five items (e.g., "My parents try to make me do things I do not want to do in my free time"). The mean score was 2.34 (*SD* = 0.95) and Cronbach's alpha = 0.89.

Boredom in leisure was developed based on past research specifically for this paper. The analytic process is described in the next section. The overall multidimensional boredom in leisure measure which included 17 items had a mean of 2.45 (*SD* = 0.68) with a Cronbach's alpha reliability score of 0.88.

### 2.3. Analyses

Based on informal focus groups and interviews with Portuguese adolescents and first-year university students, we modified an existing measure of boredom in leisure (originally a part of the Leisure Experience Battery for Adolescents [68]) to fit the cultural context. Using a combination of principal components factor analysis with varimax rotation and Cronbach's reliability analysis, we developed the measure of boredom for this paper. Table 1 includes the items in each factor, the rotated factor matrix and factor loadings for each of the four factors: nothing to do, lack of environmental stimulation, lack of autonomy, and lack of energy. Individually, these factors held together well and had acceptable to good Cronbach reliability scores; Cronbach's alphas for each factor were 0.86, 0.85, 0.80 and 0.62, respectively.

Hypothesis testing used a series of hierarchical linear regression models that predicted the WBE variables and controlled for sex, grade level, and domicile. Following the LACE conceptual framework, variables were entered in blocks beginning with person-level covariates—sex, (female or male), where the adolescent lived (central city or outside of central city), and grade level (10, 11, 12). Next, degree of active leisure was entered (number of hours in active leisure), followed by leisure variables (boredom in leisure, healthy leisure, and ability to restructure a boring situation), and then followed by a block of the parent variables (perceptions of parental knowledge, autonomy support, and autonomy control). Predicting boredom in leisure followed the same strategy with sex, grade, and domicile entered first, followed by each of the leisure-related variables. Figures 1–7 display the results of these analyses.

**Table 1.** Rotated Factor Solution for Boredom Measure.

|  | Lack of Interest & Nothing to Do | Lack of Environmental Stimulation | Lack of Autonomy | Lack of Energy |
|---|---|---|---|---|
| I have nothing to do in my free time | 0.804 | 0.214 | 0.058 | 0.003 |
| I usually don't do anything, but I don't know what to do | 0.737 | 0.210 | 0.164 | −0.075 |
| I don't have activities to fill the time | 0.697 | 0.123 | −0.081 | 0.157 |
| In my free time there is nothing interesting to do | 0.672 | 0.361 | 0.156 | 0.110 |
| I usually don't like what I am doing, but I don't know what else to do | 0.666 | 0.311 | 0.214 | 0.025 |
| I have nothing to do, but I want to do something | 0.635 | 0.354 | 0.168 | −0.079 |
| I just don't feel like doing anything in my free time | 0.601 | 0.081 | 0.167 | 0.202 |
| Free time is boring | 0.545 | −0.007 | 0.162 | 0.151 |
| The environments I'm in are usually not stimulating | 0.210 | 0.862 | 0.154 | 0.073 |
| The environments I'm in are usually not attractive | 0.251 | 0.836 | 0.146 | 0.138 |
| What I do doesn't capture my interest or engage me | 0.377 | 0.636 | 0.284 | 0.178 |
| I am not allowed to do what I want to | 0.135 | 0.067 | 0.888 | 0.062 |
| I can't do what I want to | 0.096 | 0.138 | 0.886 | 0.057 |
| I am tired of doing what I don't like in my free time | 0.184 | 0.394 | 0.617 | 0.103 |
| I usually have no control over what I'm doing | 0.392 | 0.314 | 0.449 | −0.143 |
| I have more energy than is required of me | 0.003 | −0.066 | 0.046 | −0.861 |
| I often seem to be tired and have no energy | 0.278 | 0.193 | 0.208 | 0.732 |

Extraction Method: Principal Component Analysis.
Rotation Method: Varimax with Kaiser Normalization.

Rotation converged in 5 iterations.

## 3. Results

Results from the regression analyses are presented using bar graphs that display significant beta weights for each predictor variable that is associated with each dependent variable. Results are presented in the following order: life satisfaction, subjective happiness, self-esteem, self-efficacy, positive affect, negative affect, and boredom (all dependent variables). For each regression, covariates are entered first, followed by the primary variables of interest.

### 3.1. Predicting Life Satisfaction and Subjective Happiness

The major positive predictors of life satisfaction (adjusted R-square = 0.25) were perceptions of parental autonomy support (the strongest predictor) and having healthy leisure, followed by being bored which was negatively associated with life satisfaction (see Figure 1). Being in grade 12 was also positively associated with life satisfaction.

With regard to subjective happiness (see Figure 2), data suggest that adolescents who perceive their leisure to be healthy contributed the most to their happiness, followed by whether they felt their parents knew about their leisure activities (adjusted R-square = 0.31).

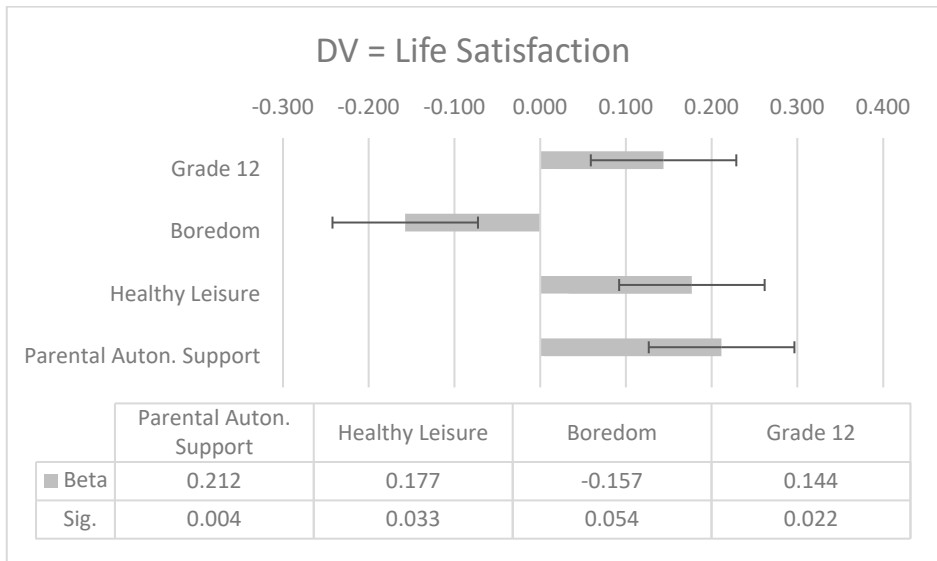

**Figure 1.** Positive predictors of life satisfaction.

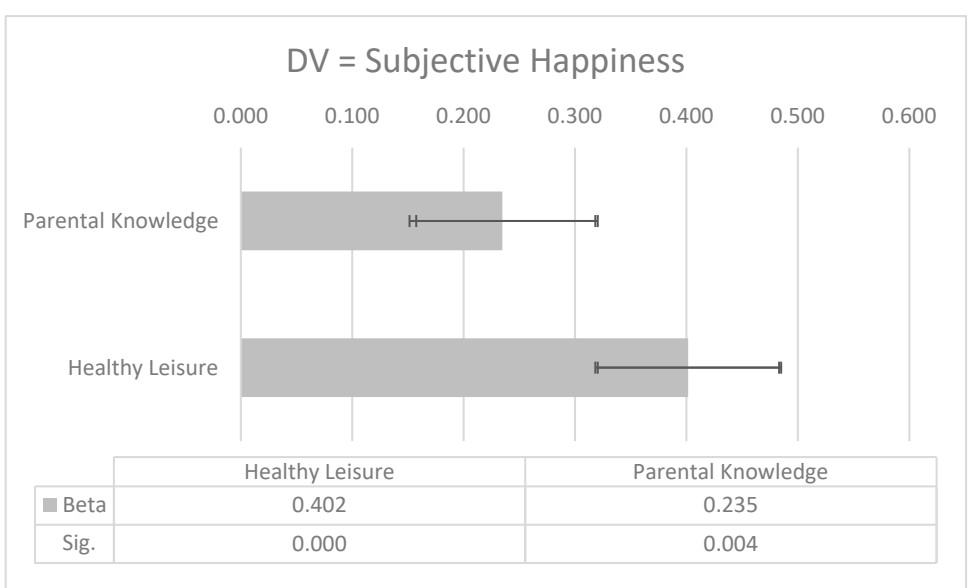

**Figure 2.** Positive predictors of subjective happiness.

### 3.2. Predicting Self-Esteem and Self-Efficacy

Positive predictors of self-esteem included being male, being in grade 12, feeling what they did in leisure was healthy, and being able to restructure a boring situation (see Figure 3). Being bored in leisure was the strongest predictor of self-esteem, (adjusted R-square = 0.36) and as hypothesized, it negatively affected adolescents' self-esteem.

Being able to restructure a boring situation was the strongest predictor of self-efficacy (adjusted R-square = 0.29). Being active in leisure also predicted self-efficacy (See Figure 4).

### 3.3. Predicting Positive and Negative Affect

Positive predictors of positive affect include being able to restructure a boring leisure situation (the strongest predictor), participating in more active activities, and having healthy leisure positively predicted positive affect (adjusted R-square = 0.36; see Figure 5).

Being bored in leisure most strongly contributed to negative affect as did being female. Being able to restructure a boring situation into something more interesting was negatively associated with negative affect (see Figure 6). The adjusted R-square was 0.16.

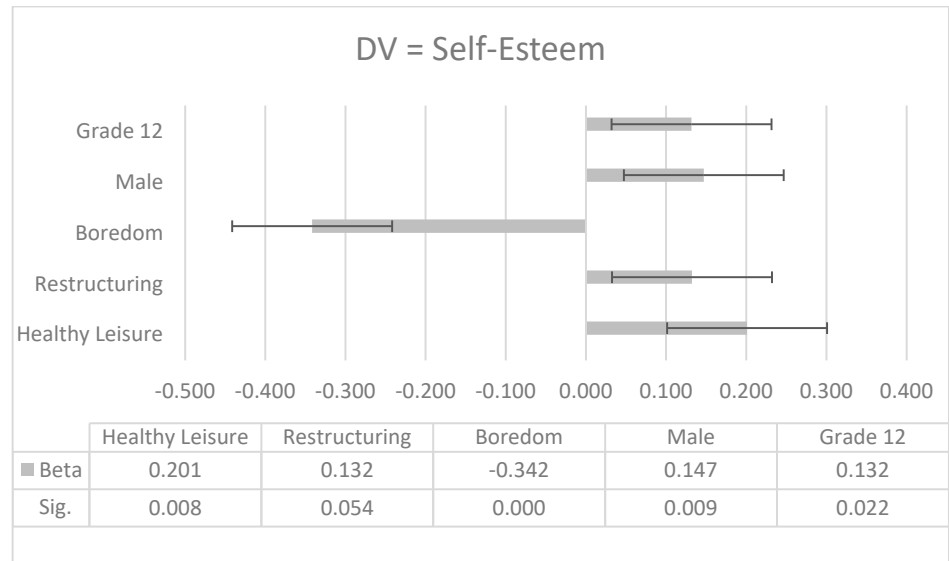

**Figure 3.** Positive predictors of self-esteem.

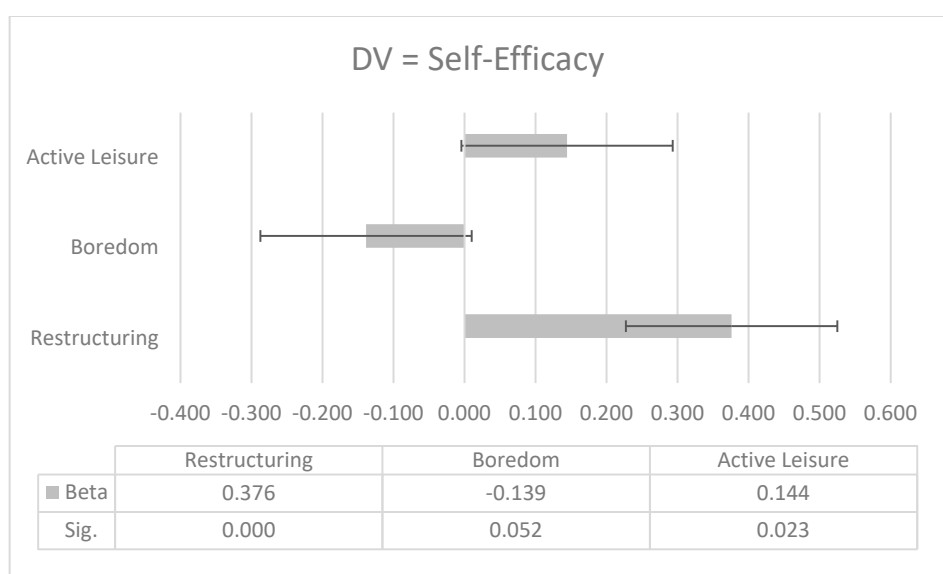

**Figure 4.** Positive predictors of self-efficacy.

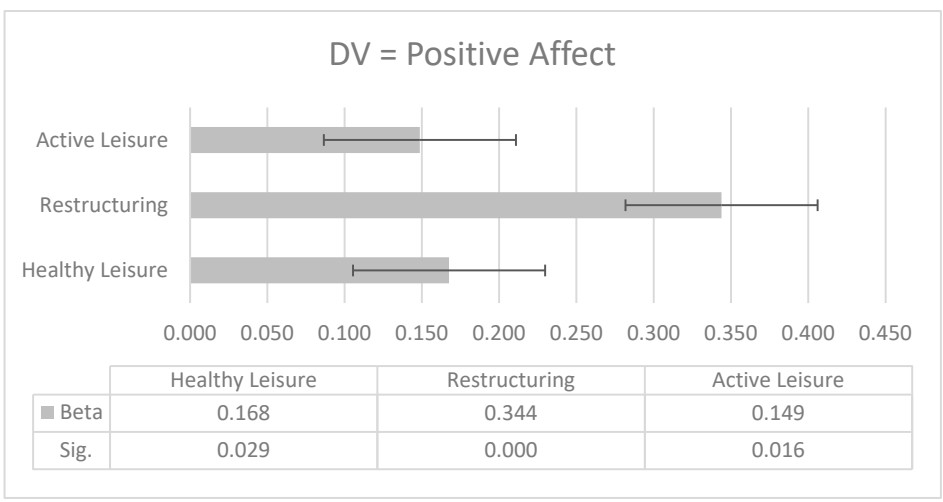

**Figure 5.** Positive predictors of positive affect.

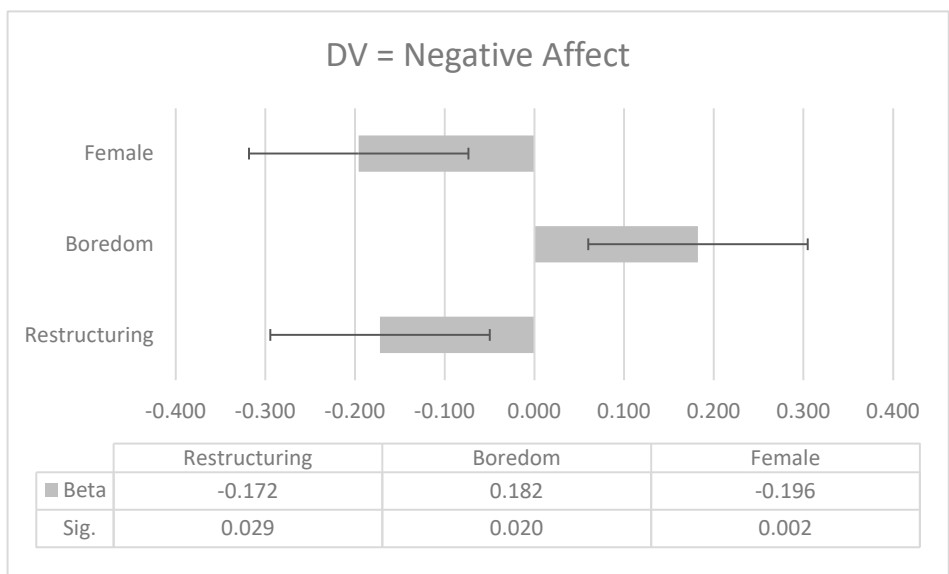

**Figure 6.** Positive predictors of negative affect.

*3.4. Predicting Boredom in Leisure*

The perception that parents have too much control of adolescent autonomy was the strongest positive predictor of boredom (see Figure 7). Having healthy leisure, on the other hand, was the strongest negative predictor of boredom, followed by being able to restructure a boring situation and being more active in leisure. Living outside of the central city also contributed to boredom. The adjusted R-square was 0.43.

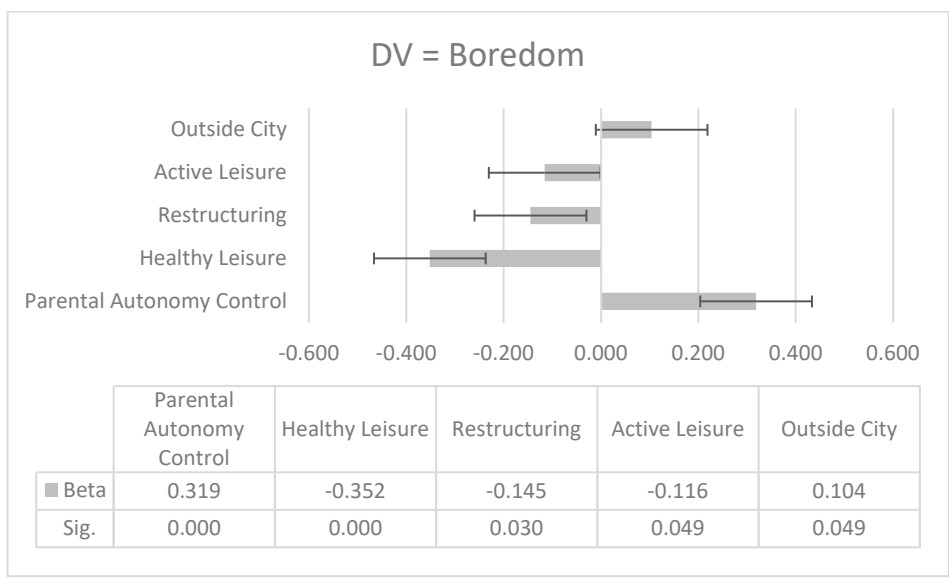

**Figure 7.** Positive predictors of boredom in leisure.

**4. Discussion**

Except for subjective happiness, experiencing boredom in leisure and/or the ability to make a boring situation more interesting were strong predictors of each of the other WBE variables in the predicted direction. Furthermore, when youth perceived what they did in leisure was healthy, they reported higher levels of life satisfaction, subjective happiness, self-esteem, and positive affect. Being active in leisure was important to adolescent self-efficacy and positive affect. Other variables that contributed to positive affect were perceiving one's leisure is healthy and being able to change a boring situation into something more interesting.

The ability to restructure a boring situation into something more interesting is not only closely connected to self efficacy but is also closely connected to initiative and persistence [69]. Being bored contributed to negative affect, while being able to restructure and being female were protective factors against negative affect. Experiencing boredom also detracted from life satisfaction, self-esteem, and self-efficacy.

Being active in leisure was not as strong of a predictor to the WBE variables as anticipated which is possibly because of the way we measured activity participation. Authors of a study of Australian high school adolescents [70] provide a discussion on measurement issues regarding activity participation and psychosocial benefits. In their study, adolescents reported spending most time in social leisure activities, fewer activities that required skill and effort, and even less involvement in structured activities. In contrast to most other studies on the importance of structured activities, they found psychosocial benefits were not associated with structured activities after controlling for other forms of leisure participation. We focused only on active leisure due to our conceptual framework hypothesizing that active engagement is more likely to be associated with WBE. In future research, a more robust conceptualization of activity measurement would be beneficial.

### 4.1. Why Is Leisure Important to WBE?

As young people transition from childhood through adolescence, they naturally have more opportunities for and crave freedom from adult control and oversight. The ability to be self determined and in control of one's actions is most easily experienced in the leisure context given that it is "supposed" to be a time of relative freedom. In this space or time, adolescents develop the capacity to direct attention and effort towards setting and achieving goals [69,71] through perseverance and initiative, a key developmental task of adolescence. Initiative is emblematic of an engaged and interested (thus, logically not bored) individual and involves the ability to act on one's environment to create enjoyable, interesting, and challenging forms of activity engagement [71]. This follows an action-in-context perspective [6] which has conceptual ties to self-efficacy and intrinsic motivation, two other hallmarks of leisure experience [72]). From a self-determination theory (e.g., [73]) perspective, when adolescents have the opportunity *and* the skills to reshape or control their environments, they thrive and have positive health and developmental outcomes.

Therefore, leisure affords the experience of self determination, identity formation, learning to persevere, and positive (e.g., fun and exciting) experiences. As suggested by our results, even boring situations can be developmentally productive and produce wellbeing if an adolescent chooses to, has the opportunity for, and has the skills to restructure the situation into something more interesting.

### 4.2. Perceptions of Parents

It is difficult to disentangle the tension between parental autonomy support, autonomy control, and parental guidance and knowledge. In part, this is likely a product of the adolescents' developmental stage as well as their personality characteristics. In part, it is likely due to the context in which one lives. Our findings, however, suggest that when adolescents perceive their parents to be autonomy controlling, they are more bored in leisure. This is worth noting given that the experience of boredom (and the ability or inability to restructure a boring situation) is linked with both protective and risk factors (e.g., [4]).

Our findings suggest that perceiving that one's parents know about things they are interested in during leisure is important to subjective happiness. However, perceptions of autonomy support were not predictive of any of the WBE experiences or less boredom. It is likely that for these Portuguese adolescents, parental knowledge was perceived as autonomy support. Given that most of the sample lived in an urban area with easy public transportation, autonomy support may not have been as important as perceiving that parents knew about their interests, what they did with their free time, and who their friends were. Other researchers in Portugal concluded that adolescents in their sample perceived their parents employed a democratic style of parenting rather than authoritarian or permissive styles [74].

Our results are in line with previous studies showing that during the transition from childhood to late adolescence, children who perceive that their behaviors are increasingly self determined and whose parents support their autonomy are more motivated [46,75]. On the other hand, children who perceive that their parents are autonomy controlling exhibit amotivation [47] and loss of self-determination [48,49]. Despite the strong cultural belief about the decrease of the importance of parental contributions compared to the input of peers during adolescence, empirical research has shown that parenting is a strong determinant of adolescent health and wellbeing during this period of life often more than peer processes [5].

The tension between parental autonomy support and autonomy control may be related to overprotection (e.g., [76], thwarting adolescents' ability to experiment, take responsibility, and learn from their actions). If parental guidance does not foster self determination, adolescents may have limited opportunities for taking initiative and learning how to restructure boring situations, thus potentially leaving youth vulnerable to boredom, amotivation, and perceiving that their leisure is not healthy.

*4.3. The Role of Boredom in Leisure*

In this study, boredom was an independent variable to predict six measures of wellbeing, and it was also a dependent variable to better understand the correlates of boredom among this sample of Portuguese youth. In considering the role of boredom, it is important to note that boredom can be a trait or a state [77–79]. That is, it can be a reaction to a specific situation (state), or it can be a personality factor that is something more chronic (trait). One study on South African adolescents (N = 2580) attempted to disentangle the roles of both state and trait boredom on substance use [41]. Using longitudinal data from three to eight biannual measurement occasions, researchers found those with higher trait boredom used more substances than those with lower trait boredom. Those who experienced higher levels of state boredom than their typical levels also reported higher levels of substance use. Other research [80] has examined how changes in an adolescent's boredom across time are associated with substance use. Among changes in other leisure experiences, adolescents who become more bored from the eighth grade to the eleventh grade were also more likely to use alcohol, smoke cigarettes, and use marijuana. Both of these studies suggest that future research on Portuguese youth would benefit from a longitudinal exploration of leisure boredom, including state and trait boredom and wellbeing.

In the present study, the one-time measure of boredom included items that related to both state and trait issues, and although it was developed through a process that supported content validity and attained a good reliability score (0.885), it was still a single measure at a single point in time. It is possible that trait boredom is more closely related to wellbeing, but it is likely that state boredom plays an important role. Other research could focus more closely and the characteristics of youth who are better able to restructure a boring situation into something more interesting. To our knowledge, there are no studies that examine the ability to restructure with state or trait boredom or other personality or contextual factors.

Although we had a limited set of variables from which to predict boredom in leisure, our findings give insight into ways to help adolescents avoid boredom while being in a leisure situation. Findings suggest providing opportunities and helping youth gain skills for healthy, active, and engaged leisure are protective factors against boredom. From a more contextual perspective, those youth who lived outside of the city seemed to experience fewer opportunities for leisure, for example. Perhaps most striking, however, was that when youth perceived their parents to be autonomy controlling, they reported higher levels of boredom. Although this could be a parental response to their child's behavior, other research has suggested this finding is driven by perceptions of parents, not parents reacting to their child's behavior [46].

*4.4. Limitations and Future Studies*

There are a number of limitations to be considered in this study. Due to considerations of participant burden, we had only a limited set of variables to examine wellbeing, leisure, and boredom in leisure. Further study in this area would benefit by including other measures of leisure participation, measures of self-determination and initiative, and a more robust set of parent variables, possibly also surveying parents directly. Participants in this study came from only two city high schools and were a homogeneous sample. Expanding the sample to include youth from rural areas would provide additional insight on adolescents' WBE. This study, one of the first to address this set of WBE variables and leisure, was cross sectional. Measuring these constructs over time, and/or using an experience sampling method, would no doubt further contribute to this line of research supported in an ecologically based perspective.

It is also important to highlight that the study is correlational, not explanatory, although the hypotheses were based on strong theoretical foundations. There were also seven hierarchical regression analyses conducted which raises the possibility of Type 1 errors. Future research should include larger sample sizes so that corrections for Type 1 errors could be utilized without loss of power to detect "true" significant values. With only 303 adolescents in this study, the loss of power to detect significant results forced a trade off by controlling for Type 1 errors using Bonferroni correctios or applying the False Discovery Rate method and the risk of making Type 11 errors. In the end, we felt comfortable taking that risk because our analyses were deliberately constructed using hierarchical linear regression in which we entered blocks of variables based on theory. Readers, however, can come to their own conclusions based on the data.

## 5. Conclusions

In general, the findings of this study suggest leisure-related variables and, to a lesser extent, adolescent perceptions of how their parents relate to them around leisure contribute to, and detract from, wellbeing experience in this sample of Portuguese adolescents. In a few cases, covariates also contributed to WBE: gender (for self-esteem and negative affect), grade level (for self-esteem and life satisfaction), and domicile (for boredom), although these were less strongly predictive than other variables.

Findings suggest the central elements of leisure—autonomy and freedom, self determination, ability to positively interact with one's environment, active engagement, and interest—contribute to wellbeing experience. The question of what contributes to subjective happiness and general wellbeing seems to be, in part, an existential response to one's situation, as well as the ability to exert some semblance of control over one's situation and not be controlled by external forces. In a leisure situation, at least, adolescent perceptions of having personal control are essential, underscoring the importance of developing skills related to initiative, persistence, and the ability to restructure a "boring" situation into something that provides interest and challenge, all of which contribute to wellbeing experience.

From a practical perspective, a few suggestions arise from this study that may promote adolescent wellbeing from a bioecological perspective. These suggestions are in line with the Portuguese National Mental Health Plan (PNSM [55]) and also offer an under-utilized context to promote mental health and wellbeing: leisure. To begin, three main groups could be the targets for programs to educate about the importance of healthy use of leisure time to their wellbeing: parents, teachers, and youth. Parents play a strong role in their children's leisure activities, and this role is likely more pronounced in Portugal, given the integral importance of family. Although often considered of trivial developmental importance, as noted, leisure is an important context for health promotion and risk prevention. Equipping parents with knowledge and strategies on how to promote and support healthy leisure may serve to increase their adolescent's wellbeing. Similarly, teachers can also include lessons and opportunities for adolescents to explore leisure interests and find passions that may carry through adulthood. Both efforts would help adolescents themselves learn that leisure is an

important part of life and wellbeing. Furthermore, they would learn specific knowledge and skills that would enable them to practice healthy leisure and avoid boredom.

Structurally, municipal and rural policy makers may want to inventory the opportunities and resources that exist in the local environment. Given the youth in non-urban areas reported being more bored than their urban counterparts, increasing opportunities in the local context may contribute to wellbeing and development.

**Author Contributions:** Conceptualization, L.C. and T.F.; Methodology, L.C. and T.F.; Investigation, L.C. and T.F.; Writing—original draft, L.C.; Writing—review & editing, T.F. All authors have read and agreed to the published version of the manuscript.

**Funding:** This study was conducted at the Psychology Research Centre (PSI/01662), School of Psychology, University of Minho, supported by the Foundation for Science and Technology (FCT) through the Portuguese State Budget (UIDB/PSI/01662/2020).

**Institutional Review Board Statement:** The study was conducted in accordance with the Declaration of Helsinki, and approved by the Ethics Committee for Research in Social and Human Sciences (CEICSH) of the University of Minho (CEICSH 046/2019; 11 October 2019).

**Informed Consent Statement:** Informed consent was obtained from all subjects involved in the study.

**Data Availability Statement:** We are happy to provide the data which are available by request to the second author.

**Acknowledgments:** The authors would like to acknowledge the Portuguese Fulbright Commission for their financial support for the first author for living in the country for three months.

**Conflicts of Interest:** The funders had no role in the design of the study; in the collection, analyses, or interpretation of data; in the writing of the manuscript; or in the decision to publish the results.

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
