# Peer review of "Understanding the Role of Leisure in Portuguese Adolescent Wellbeing Experience"

_2673-995X, doi:10.3390/youth3020041_

Round 1
Reviewer 1 Report
Thank you for the opportunity to review this interesting research titled “Portuguese Adolescent Wellbeing Experience: Understanding the Roles of Leisure and Boredom”. Authors investigated the role of leisure and leisure boredom in Portuguese adolescent wellbeing, and predictors of boredom in leisure among this population using a sample of 303 adolescents in grades 10, 11, and 12 living in a large urban area in northern Portugal.
Overall, the study is well organized and comprehensively described, and is expected to have a significant contribution to our knowledge on adolescent health.
However, this reviewer has some concerns and comments:
1- Contextual information are much needed. This study was conducted in one urban area in northern Portugal. It is very important for readers and researchers to understand the context of this area and characteristics of its population. This can also help us gauge the generalizability of the findings. This should be also be reflected in the study model as ‘context’ is already a major part of it.
2- The background section includes extremely limited information on Portuguese Adolescent health. While there might be no studies that investigated the specific variables used in this study, a simple search can reveal plenty of evidence on this population psychosocial health. There are also tens on studies that considered adolescent-parent issues in Portugal. A synthesis of such evidence is needed.
3- Please include a clear purpose statement and/or specific aims.
4- Please include a separate section that defines the study primary as well as secondary variables. In same section also please clarify the independent and dependent variables and covariates.
5- What are the specific culturally competent implications of the study's findings for education, research, healthcare providers, and policy makers? Kindly elaborate on this while avoiding general statements and clichés.
Thank you!
Author Response
The authors would like to thank Reviewer 1 for the very helpful comments. We have done our best to revise the manuscript to improve it based on the comments.
Thank you for the opportunity to review this interesting research titled “Portuguese Adolescent Wellbeing Experience: Understanding the Roles of Leisure and Boredom”. Authors investigated the role of leisure and leisure boredom in Portuguese adolescent wellbeing, and predictors of boredom in leisure among this population using a sample of 303 adolescents in grades 10, 11, and 12 living in a large urban area in northern Portugal.
Overall, the study is well organized and comprehensively described, and is expected to have a significant contribution to our knowledge on adolescent health.
However, this reviewer has some concerns and comments:
- Contextual information are much needed. This study was conducted in one urban area in northern Portugal. It is very important for readers and researchers to understand the context of this area and characteristics of its population. This can also help us gauge the generalizability of the findings. This should be also be reflected in the study model as ‘context’ is already a major part of it.
Response: Thank you for this request. We agree that adding information about the context will help the reader better understand the results as well as issues of generalizability.
- The background section includes extremely limited information on Portuguese Adolescent health. While there might be no studies that investigated the specific variables used in this study, a simple search can reveal plenty of evidence on this population psychosocial health. There are also tens on studies that considered adolescent-parent issues in Portugal. A synthesis of such evidence is needed.
Response: We have added additional information on Portuguese adolescent health.
- Please include a clear purpose statement and/or specific aims.
Response: We have re-written the purpose statement and hypotheses and hope that these are more clearly stated.
- Please include a separate section that defines the study primary as well as secondary variables. In same section also please clarify the independent and dependent variables and covariates.
Response: We hope we have made the distinction among types of variables (independent, dependent, and covariate) clearer.
- What are the specific culturally competent implications of the study's findings for education, research, healthcare providers, and policy makers? Kindly elaborate on this while avoiding general statements and cliché
Response: Thank you for this suggestion. We added a paragraph to the conclusions on implications for teachers, parents, adolescents, and policy makers.
Reviewer 2 Report
Thank you for the opportunity to review your manuscript. I felt that this manuscript was well-written and interesting. Further, it begins to fill some gaps in the boredom literature around boredom in the family system with the parental autonomy analyses. I have a few comments and suggestions to further improve your manuscript.
1. Results section - It would be useful to add a brief paragraph at the beginning of the results section to orient the reader to how your plan to present your results. You have some of that in the previous section, but it would be helpful for your reader to be able to pick up in the results section with this orientation.
2. Results section - Be consistent in your reporting of your R-squared values. Some places you report as percent variance explained, some places its adjusted R-squared.
3. Results section - There are several analyses in this manuscript. Please address how you are dealing with the multiple comparisons issue which might lead to false positives. If there are not corrections already applied, these analyses need to be reconsidered with some control for multiple comparisons.
4. Discussion section - It would be great to see some discussion about the correlational nature of these results. Some people experience boredom much more frequently than others (i.e., trait boredom), and since boredom is known to reduce motivation it may be that these individuals have a much harder time restructuring their time for active leisure due to this issue. A careful consideration of the correlational nature of each of these findings would add to the contribution of this manuscript.
Author Response
Thank you for these very helpful and thoughtful comments. We have done our best to improve the manuscript based on the comments.
Thank you for the opportunity to review your manuscript. I felt that this manuscript was well-written and interesting. Further, it begins to fill some gaps in the boredom literature around boredom in the family system with the parental autonomy analyses. I have a few comments and suggestions to further improve your manuscript.
- Results section - It would be useful to add a brief paragraph at the beginning of the results section to orient the reader to how your plan to present your results. You have some of that in the previous section, but it would be helpful for your reader to be able to pick up in the results section with this orientation.
Response: Thank you, we have added a brief paragraph that hopefully provides a clearer orientation.
- Results section - Be consistent in your reporting of your R-squared values. Some places you report as percent variance explained, some places its adjusted R-squared.
Response: Thank you for picking that up. It has been changed.
- Results section - There are several analyses in this manuscript. Please address how you are dealing with the multiple comparisons issue which might lead to false positives. If there are not corrections already applied, these analyses need to be reconsidered with some control for multiple comparisons.
Response: This is an excellent point and we generally agree, although we were unable to do corrections due to the loss of power due to the small sample size as this was a small exploratory study. Below is what we wrote. Hopefully this clearly makes the point and also explains why we were unable to correct. Doing a Bonferroni or FDR method practically eliminated all significant results. Based on my extensive prior research with the leisure variable on huge data sets (e.g., N = 5,000 to 10,000), and the strong theoretical foundation, we hope that our perspective is reasonable. I’m not sure how or why we omitted the significance levels of the beta weights presented, but they have now been included.
We considered providing all of the tables from the hierarchical regression equations but believe that this more simplistic way of viewing the results is easier to digest.
We added:
It is also important to highlight that the study is correlational, not explanatory, although hypotheses were based on strong theoretical foundations. There were also seven hierarchical regression analyses conducted, which raises the possibility of Type 1 errors. Future research should include larger sample sizes so that corrections for Type 1 errors could be utilized without loss of power to detect “true” significant values. With only 303 adolescents in this study, the loss of power to detect significant results forced a trade-off by controlling for Type 1 error using Bonferroni correctios or applying the False Discovery Rate method and the risk of making Type 11 errors. In the end we felt comfortable taking that risk because our analyses were deliberately constructed using hierarchical linear regression in which we entered blocks of variables based on theory. Readers, however, can come to their own conclusions based on the data.
- Discussion section - It would be great to see some discussion about the correlational nature of these results. Some people experience boredom much more frequently than others (i.e., trait boredom), and since boredom is known to reduce motivation it may be that these individuals have a much harder time restructuring their time for active leisure due to this issue. A careful consideration of the correlational nature of each of these findings would add to the contribution of this manuscript.
Response: These are great suggestions and we hope we have done them justice. We did not address the correlational nature of the findings for each of the findings, but rather wove into the manuscript a stronger discussion and caveats given the correlational nature of the data.
We also appreciate the suggestion to address state and trait boredom, and added stronger recommendations for future studies to address state and trait boredom, as well as change over time, and added a couple of references to support those recommendations.
Round 2
Reviewer 1 Report
This revised version is much improved. Authors have responded to all comments appropriately. I have no further concerns.